# Cooperative regulation by G proteins and Na⁺ of neuronal GIRK2 K⁺ channels

**Weiwei Wang[1], Kouki K Touhara[1], Keiko Weir[2], Bruce P Bean[2]\*, Roderick MacKinnon[1]\***

[1]Laboratory of Molecular Neurobiology and Biophysics, Howard Hughes Medical Institute, Rockefeller University, New York, United States; [2]Department of Neurobiology, Harvard Medical School, Boston, United States

**Abstract** G protein gated inward rectifier K⁺ (GIRK) channels open and thereby silence cellular electrical activity when inhibitory G protein coupled receptors (GPCRs) are stimulated. Here we describe an assay to measure neuronal GIRK2 activity as a function of membrane-anchored G protein concentration. Using this assay we show that four Gβγ subunits bind cooperatively to open GIRK2, and that intracellular Na⁺ – which enters neurons during action potentials – further amplifies opening mostly by increasing Gβγ affinity. A Na⁺ amplification function is characterized and used to estimate the concentration of Gβγ subunits that appear in the membrane of mouse dopamine neurons when GABA$_B$ receptors are stimulated. We conclude that GIRK2, through its dual responsiveness to Gβγ and Na⁺, mediates a form of neuronal inhibition that is amplifiable in the setting of excess electrical activity.

**\*For correspondence:**
bruce_bean@hms.harvard.edu
(BPB); mackinn@rockefeller.edu
(RM)

**Competing interests:** The authors declare that no competing interests exist.

## Introduction

Potassium channels oppose membrane electrical excitability by driving the membrane voltage towards the K⁺ reversal potential, near -90 mV in mammalian neurons. In the nervous system many inhibitory neurotransmitters act through G protein coupled receptors (GPCRs), which regulate a G protein gated inward rectifier K⁺ (GIRK) channel (*Pfaffinger et al., 1985*; *Lesage et al., 1994*; *Lesage et al., 1995*; *Wang et al., 2014*). In this form of signaling G proteins are released by stimulated GPCRs and diffuse on the cytosolic surface of the membrane to a site on the K⁺ channel. A tight complex of the β and γ G protein subunits (known as the 'Gβγ subunit') binds to GIRK, favors the open conformation, and drives the membrane potential towards the K⁺ reversal potential (*Pfaffinger et al., 1985*; *Logothetis et al., 1987*; *Reuveny et al., 1994*; *Krapivinsky et al., 1995*) (*Figure 1A*).

Extensive research on GIRK channels in both neurons and cardiac cells has identified three important regulators of GIRK channel gating: Gβγ subunits, the signaling lipid PIP$_2$ and intracellular sodium (*Logothetis et al., 1987*; *Wickman et al., 1994*; *Huang et al., 1998*; *Sui et al., 1998*; *Ho and Murrell-Lagnado, 1999a*; *Petit-Jacques et al., 1999*). Many aspects of how these ligands interact with the channel, whether they are absolutely required for channel opening, and how they interact with each other through their respective interactions with the channel have remained unclear. Some studies supported an absolute requirement for Gβγ subunits (*Logothetis et al., 1987*; *Wickman et al., 1994*; *Krapivinsky et al., 1995*), while others concluded that Mg²⁺-ATP with Na⁺ (*Petit-Jacques et al., 1999*) or PIP$_2$ (*Huang et al., 1998*) were by themselves sufficient to open GIRK channels in the absence of Gβγ subunits. Furthermore, attempts to determine Gβγ affinity for GIRK channels in cell membranes, assessed by applying detergent-solubilized Gβγ to membrane patches, yielded values ranging from 3 nM to 125 nM (*Wickman et al., 1994*; *Krapivinsky et al., 1995*). The problem with these studies is the membrane partition coefficient for detergent-solubilized Gβγ is

**eLife digest** Signals from outside of a cell can alter the activity inside the cell. This process often involves members of a large family of proteins called G protein-coupled receptors (GPCRs) that are found on the surface of many cells in the body. When these receptors are activated they release a G protein on the inside of the cell that then splits into two parts. One of these parts – called the Gβγ subunit – can directly bind to, and open, a protein channel called a GIRK channel in the cell membrane. Once opened, these channels allow potassium ions to flow into the cell.

GIRK channels are involved in a number of processes in the body. For example, GIRK2 is a major type of GIRK channel found in nerve cells. When this channel is activated the flow of potassium ions into the cell inhibits the nerve cell's activity and makes it less likely to send electrical impulses. However, it was not clear how many Gβγ subunits are required to activate a GIRK2 channel.

Now, Wang et al. report that four Gβγ subunits must bind to a GIRK2 channel and then work together to open it. This means that a GIRK2 channel will switch between a closed and an open state whenever the density of Gβγ subunits released onto the cell membrane reaches a certain threshold.

Wang et al. also found that a high concentration of sodium ions in the cell causes the Gβγ subunits to bind more strongly to the GIRK2 channel; this makes that channel more likely to open and inhibit the nerve cell's activity. This action serves to dampen down the activity of the most active neurons, because highly active nerve cells contain more sodium. Also, in a related study, Touhara et al. – who include many of the same researchers – discovered that sodium ions affect GIRK4 channels from heart cells in a similar way.

These findings shed new light on G protein signaling, but there is still more that is not yet completely understood. Wang et al.'s findings suggest that the concentration of Gβγ subunits in certain nerve cells is much higher than previously expected, and further work is now needed to explore how this might be achieved.

not known and therefore the membrane concentration is not known. Isothermal titration calorimetry (ITC) measurements with a soluble form of Gβγ (lipid anchor-removed) and a soluble cytoplasmic domain of a GIRK channel (removed from the transmembrane pore) measured the affinity to be 250 μM (*Yokogawa et al., 2011*). These experiments can only report binding affinity in the absence of energetic coupling to a gated pore, which was absent in the experiment.

Given the difficulty in knowing accurately the composition and quantity of components in living cell membranes – the experimental system in which the majority of studies had been carried out – we developed a total reconstitution assay to investigate the regulation of neuronal GIRK2 channel gating (*Wang et al., 2014*). Using planar lipid bilayers in which purified GIRK2 channels, G protein subunits and $PIP_2$ were reconstituted, we found that Gβγ subunits and $PIP_2$ are simultaneously required to open GIRK2 channels (each alone is insufficient) and that $Na^+$ is not required for opening, but modulates GIRK2 channel opening. Because planar bilayers allow quantitative control of lipid concentrations, the reconstitution study also permitted a detailed characterization of channel opening as a function of the $PIP_2$ concentration.

While it is possible to specify lipid (e.g. $PIP_2$) concentrations in a planar bilayer membrane, it is not possible to specify protein concentrations by simply mixing components during the bilayer membrane synthesis. For this reason, the reconstitution study described above did not permit accurate control of membrane Gβγ concentration. In the current study we present a method to specify Gβγ subunit concentration in planar lipid membranes and use the method to determine the Gβγ-GIRK2 channel activity relationship. We then show that intracellular $Na^+$ regulates GIRK2 channel gating mostly by increasing the GIRK2 affinity for Gβγ. Finally, we use the newly defined quantitative relationship between Gβγ, $Na^+$, and GIRK2 channel activity to estimate the membrane concentration of Gβγ subunits that appear in mouse dopamine neuron membranes upon stimulation of $GABA_B$ receptors.

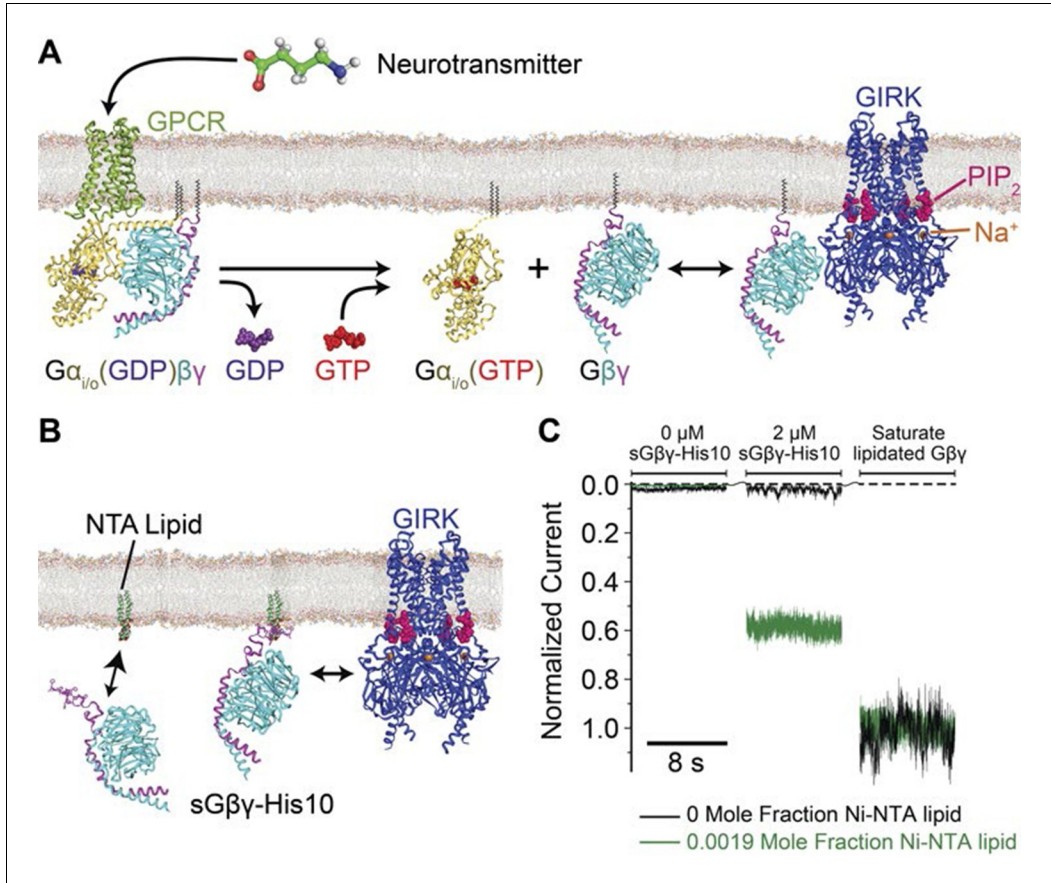

**Figure 1.** Membrane anchored Gβγ binds to GIRK and activates the channel. (**A**) Inhibitory neurotransmitters activate $G_{i/o}$ G protein coupled receptors (GPCRs) in neuron membranes. The GPCRs facilitate the exchange of GDP to GTP on the G protein hetero-trimer, releasing the $G\alpha_{i/o}$ subunit and Gβγ subunit. The membrane-anchored Gβγ subunit binds to and activates GIRK. (**B**) NTA lipid (head group modified with a $Ni^{2+}$ chelator NTA, DOGS-NTA) is used to anchor non-lipid modified and His-tagged Gβγ (sGβγ-His10) onto the lipid membranes. 2 μM of sGβγ-His10 was used to fully saturate all NTA lipid on the membrane. The sGβγ-His10 density on the membrane can be controlled by the NTA lipid mole fraction. 32 μM C8-PIP2 was included on the same side as sGβγ-His10. (**C**) Membrane-bound sGβγ-His10 activates GIRK to different levels depending on the NTA lipid mole fraction. Lipid modified Gβγ is used to fully activate GIRK at the end of each experiment. GIRK currents corresponding to different NTA lipid mole fractions are normalized to the fully activated value. A detailed description of the experiment is shown in *Figure 1—figure supplement 1*. Example current traces of activation by sGβγ-His10 and lipid modified Gβγ in liposomes are shown in *Figure 1—figure supplement 2*.

The following figure supplements are available for figure 1:

**Figure supplement 1.** Details of the planar bilayer experiment.

**Figure supplement 2.** Example traces of GIRK2 activation by sGβγ-His10 and lipid modified Gβγ in proteoliposomes.

## Results and discussion

### Controlling membrane G protein concentration

A method to control the concentration of G proteins on the surface of a lipid bilayer membrane is illustrated (*Figure 1B*, *Figure 1—figure supplement 1*). GIRK2 channels were reconstituted into planar lipid membranes formed with known mole fractions of Ni-NTA lipid, doped into otherwise biological phospholipids (*Nye and Groves, 2008*; *Knecht et al., 2009*; *Platt et al., 2010*; *Masek et al., 2011*). Modified Gβγ subunits with a His-tag replacing the lipid anchor on the γ subunit were then

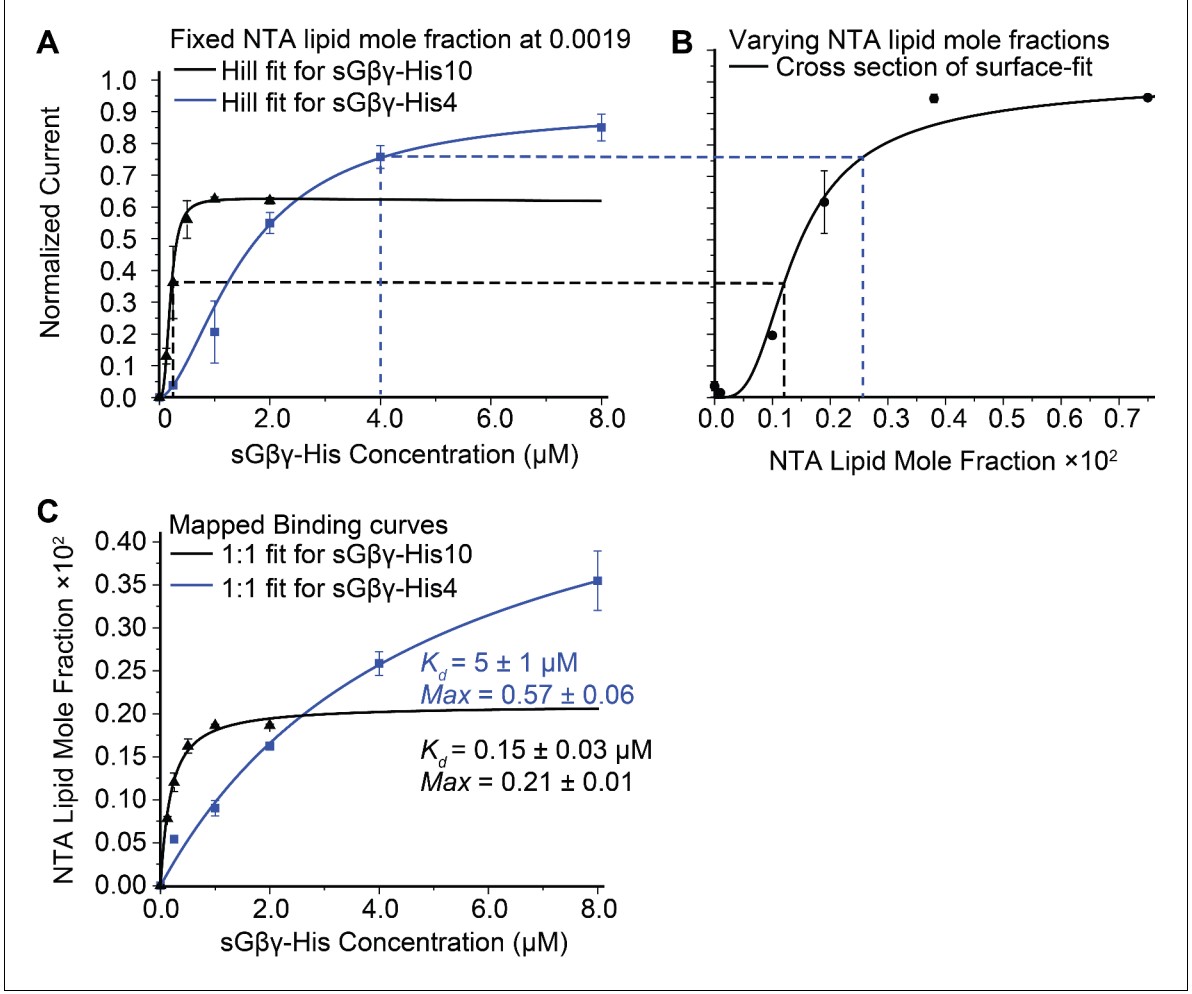

**Figure 2.** Calibration of sGβγ-His10 binding to NTA lipid. (**A**) Titration of GIRK activity by sGβγ-His10 (black) or sGβγ-His4 (blue) to lipid membranes containing a fixed 0.0019 mol fraction of NTA lipid (n = 3–5 membranes, mean ± SEM). Solid lines are fits to the Hill equation: $Current = Max \times [G\beta\gamma]^n/(K_d^n + [G\beta\gamma]^n)$ with $K_d$ = 200 ± 15 nM, n = 2.9 ± 0.4 (black, sGβγ-His10) and $K_d$ = 1.6 ± 0.1 μM, n = 1.7 ± 0.05 (blue, sGβγ-His4). Solutions contained 150 mM KCl on both sides of the membrane and 32 mM NaCl on the inside where sGβγ-His was applied. (**B**) Titration of GIRK activity as a function of NTA lipid mole fraction in the presence of 2 μM sGβγ-His10 in solution. The solid line corresponds to a model (*Figure 3—figure supplement 1*). (**C**) Mapping of the NTA lipid concentration in panel (**B**) to the solution sGβγ-His concentration in panel (**A**) through GIRK activity. The curves are rectangular hyperbolas (Hill equation with n = 1) with $K_d$ = 0.15 ± 0.03 μM (black) and $K_d$ = 5.0 ± 1.0 μM (blue). A similar affinity ($K_d$ = 0.09 ± 0.02 μM) was obtained for fluorescently labeled sGβγ-His10 adsorption to giant unilamellar vesicles (GUVs) containing 0.03 mol fraction of NTA lipid (*Figure 2—figure supplement 1*).

The following figure supplement is available for figure 2:

**Figure supplement 1.** Binding of fluorescently labeled sGβγ-His10 to GUVs containing 0.03 mol fraction of NTA lipid.

added at known concentrations to the solution on one side of the membrane with the idea that these would anchor to the membrane via the Ni-NTA lipid (*Kubalek et al., 1994*; *Schmitt et al., 1994*; *Knecht et al., 2009*; *Platt et al., 2010*). All experiments were carried out in the presence of a fixed concentration of 32 μM C8-PIP₂ to ensure high occupation of PIP₂ sites on the channel: 32 μM C8-PIP₂, based on channel activity measurements, corresponds to 0.02 mol fraction (2%) membrane PIP₂ (*Wang et al., 2014*). Example data using this assay are shown (*Figure 1C*). In the absence of Ni-NTA lipid, addition of soluble Gβγ with a His-10 tag (sGβγ-His10) to a solution concentration of 2 μM failed to activate the channel. The presence of channels in the membrane was subsequently confirmed by addition of a maximally effective (but unknown) concentration of lipid-anchored Gβγ through vesicle fusion with the membrane. In another experiment, when the same concentration of

sGβγ-His10 was added to a membrane formed with 0.0019 mol fraction Ni-NTA (19 out of 10,000 lipid molecules in the membrane containing the Ni-NTA head group), channels were activated (*Figure 1C*). Subsequent addition of excess lipid-anchored Gβγ to the same membrane showed that about 60% of the GIRK channels had been activated by the Ni-NTA lipid-anchored Gβγ. All further experiments were performed in the manner described, ending with saturation of the membrane with Gβγ to achieve maximal activation of the GIRK channels present. This normalization step enables comparison of currents measured in different membranes with different numbers of GIRK channels by placing them on a common scale (normalized current).

In the assay two equilibrium reactions occur, as depicted (*Figure 1B*). First, sGβγ-His10 binds from solution to the Ni-NTA lipid, and second, the sGβγ-His10-Ni-NTA lipid complex binds to the channel. We are ultimately interested in the second reaction as this determines channel activation as a function of Gβγ concentration on the membrane (Gβγ density in 2 dimensions). In *Figure 2A* the black symbols and curve show the normalized GIRK current level as a function of sGβγ-His10 solution concentration with a membrane containing Ni-NTA lipid at a mole fraction 0.0019. Normalized currents under these conditions reach a maximum value around 0.6 (60% of current that is reached when the same membranes are saturated with lipid-anchored Gβγ). A maximum, saturated value below 1.0 can be explained if 2 μM sGβγ-His10 is sufficient to occupy all Ni-NTA lipid molecules in the membrane, but the concentration of Ni-NTA lipid in the membrane is too low to occupy all sites on the channel. This explanation is supported by the graph on the right (*Figure 2B*) in which normalized current is plotted as a function of Ni-NTA lipid mole fraction in the presence of 2 μM sGβγ-His10 (i.e. a concentration that is sufficient to occupy all Ni-NTA lipid molecules). This graph is asymptotic to ~1 at higher values of Ni-NTA lipid mole fraction, and, as one would expect, 0.6 on the Y-axis corresponds to 0.0019 on the X-axis. A third graph (*Figure 2C*) of values from the X-axis in *Figure 2B*, plotted as a function of corresponding values (dashed lines) from the X-axis in *Figure 2A*, isolates the binding reaction of sGβγ-His10 to Ni-NTA lipid. The curve is a rectangular hyperbola (binding isotherm) with a $K_d$ of 150 nM (*Figure 2C*, black curve). A similar binding curve and affinity were determined for fluorescent sGβγ-His10 adsorption onto giant unilamellar vesicles (GUVs) containing Ni-NTA lipid at a mole fraction of 0.03 (*Figure 2—figure supplement 1*).

The blue data points and curve in *Figure 2A* show a similar set of experiments using sGβγ-His4, that is, a soluble form of Gβγ with 4 instead of 10 histidine residues in its tag. The normalized current level is asymptotic to a value higher than 0.6 (blue curve). The explanation for this becomes evident when the (apparent) Ni-NTA lipid mole fraction is plotted against the corresponding (blue dashed lines) sGβγ-His concentration (*Figure 2C*, blue symbols and curve): in this binding isotherm the affinity is lower and the maximum apparent Ni-NTA lipid mole fraction is ~3 times higher. This result follows if sGβγ-His4 binds to a single Ni-NTA lipid and sGβγ-His10 binds to 3 Ni-NTA lipids. This stoichiometric difference is consistent with known structures of Ni-NTA-polyhistidine complexes, which show that a single Ni-NTA group is coordinated by 2 histidine residues separated by at least one histidine residue (*Knecht et al., 2009*). Thus, sGβγ-His4 can only attach to a single Ni-NTA lipid while sGβγ-His10 can - and does - attach to three.

All further experiments were carried out using sGβγ-His10 at 2 μM concentration to fully occupy the Ni-NTA lipid binding sites in the membrane and thus ensure that the concentration of Gβγ in the membrane would be controlled solely by the Ni-NTA lipid mole fraction. In other words, this approach isolates the reaction of interest – channel activity (related to occupancy in a manner to be determined) as a function of known membrane Gβγ concentration. The graphs report Gβγ concentration as Ni-NTA lipid mole fraction, while keeping in mind that the molar density of Gβγ in the membrane is one third that of the Ni-NTA lipid density.

## G protein and Na⁺ regulation of the GIRK2 channel

Having established an assay to control the concentration of Gβγ in the membrane, we measured the activity of GIRK2 as a function of membrane Gβγ concentration as well as solution Na⁺ concentration (*Figure 3A*). At each Na⁺ concentration, normalized current increases as a steep sigmoidal function of membrane Gβγ concentration (*Figure 3B*). The slope of these functions on a log-log plot at sufficiently low Gβγ concentrations (achieved in these experiments for the Gβγ titrations at lower Na⁺ concentrations) are consistent with four Gβγ subunits being required to open a GIRK2 channel (see methods) (*Figure 3C*). A strong effect of Na⁺ on the functional relationship is clear and noteworthy because in cells Na⁺ is known to regulate GIRK currents, but by a mechanism that is unknown

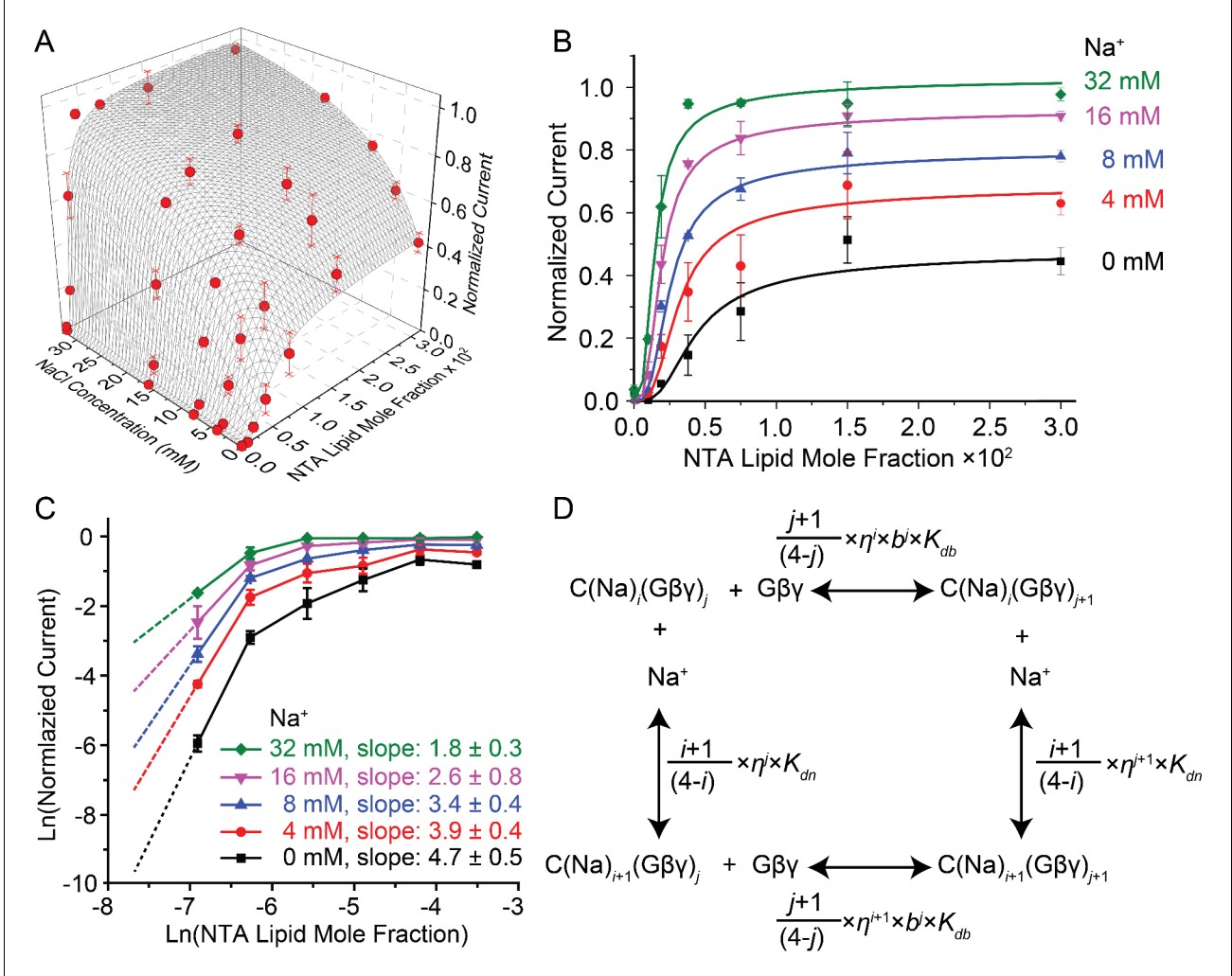

**Figure 3.** GIRK activity as a function of Gβγ and Na$^+$ concentration. 2 μM sGβγ-His10 was included in the solution on the intracellular side of GIRK. (**A**) Normalized GIRK current (red spheres, mean ± SEM, n = 3–5 membranes) is graphed as a function of Gβγ and Na$^+$ concentration. Surface mesh shows predictions of a model for ligand activation (***Figure 3—figure supplement 1***). (**B**) Data points in (**A**) are graphed as a family of curves (surface intersections) corresponding to each Na$^+$ concentration. (**C**) Log-log plot of normalized current against NTA lipid mole fraction. Data points corresponding to 0.0001 NTA lipid mole fraction were excluded because the current levels (<0.02 normalized current) were much smaller than background noise. Other data points are connected with solid lines. Dashed lines show the slope of the line connecting the first two graphed data points. (**D**) A schematic of the ligand activation model fit to the data. $i$ and $j$ are integers between 0 and 4. The fitted parameters are: equilibrium dissociation constant for the first Na$^+$ to bind in the absence of Gβγ, $K_{dn}$ = 60 ± 20 mM, equilibrium dissociation constant for the first Gβγ in the absence of Na$^+$, $K_{db}$ = 0.019 ± 0.007, cooperativity factor for each successive Gβγ binding $b$ = 0.30 ± 0.06, cross-cooperativity factor between Gβγ and Na$^+$ binding $\eta$ = 0.63 ± 0.04 and an activity term $\theta$ as described in ***Figure 3—figure supplement 1***. A comparison of fits to the data using cooperative and non cooperative models is shown in ***Figure 3—figure supplement 2***.

The following figure supplements are available for figure 3:

**Figure supplement 1.** Modeling of Na+ and Gβγ binding equilibrium with the GIRK2 tetramer.

**Figure supplement 2.** Comparison of fitted model allowing or not allowing cooperativity in Gβγ binding.

(***Sui et al., 1996***; ***Ho and Murrell-Lagnado, 1999b***; ***Petit-Jacques et al., 1999***). The titrations show that Gβγ activates the channel to a greater extent, especially at lower Gβγ concentrations, as Na$^+$ is increased (***Figure 3B***). To further understand how these two ligands interact with the channel to regulate its gating we constructed an equilibrium model. This model was guided by atomic structures, which show that a tetramer GIRK2 channel has 4 structurally identical Gβγ binding sites and 4 Na$^+$

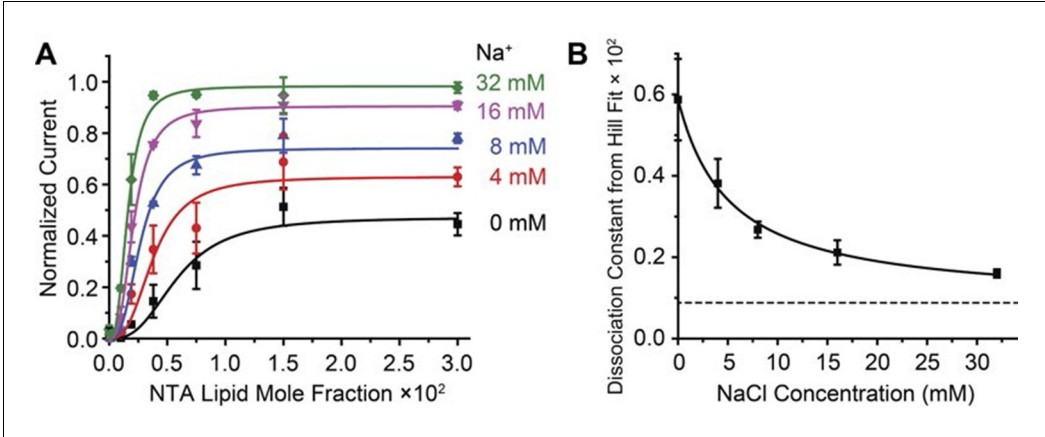

**Figure 4.** $Na^+$ concentration regulates $G\beta\gamma$ affinity. 2 μM sG$\beta\gamma$-His10 was included in the solution on the intracellular side of GIRK. (**A**) Normalized current values from *Figure 3A* are fit to the Hill equation: *Current = Max* $\times$ [NTA-lipid mole fraction]$^n$ / $(K_d^n$ + [NTA-lipid mole fraction]$^n)$ with a Hill coefficient ($n$) of 2.8 ± 0.2 for all curves and equilibrium dissociation constant ($K_d \times 10^2$) for $G\beta\gamma$ binding 0.6 ± 0.2, 0.38 ± 0.06, 0.27 ± 0.02, 0.21 ± 0.03 and 0.16 ± 0.01 for $Na^+$ concentrations 0 mM, 4 mM, 8 mM, 16 mM and 32 mM, respectively. (**B**) $G\beta\gamma$ $K_d$ values from fits in (**A**) are plotted as function of $Na^+$ concentration. The curve is the rectangular hyperbola $K_d = K_{dmax} + (K_{dmin} - K_{dmax}) \times [Na^+] / (K_{d-Na+} + [Na^+])$, where $K_{d-Na+}$ = 5.1 ± 0.9 mM is the apparent $Na^+$ dissociation constant as estimated through its effect on the affinity of $G\beta\gamma$.

binding sites (*Whorton and MacKinnon, 2013*). The model contains 25 states, corresponding to an order-independent occupation number 0 to 4 for each ligand (*Figure 3D* and *Figure 3—figure supplement 1*). Fitted parameters in the model include a dissociation constant and cooperativity factor for each ligand, a cross cooperativity factor between $G\beta\gamma$ and $Na^+$ and a parameter relating ligand occupancy to channel activity (see *Figure 3—figure supplement 1*). The data and modeling support the following conclusions. First, all four $G\beta\gamma$ binding sites must be occupied on the channel before it opens, consistent with the limiting slope analysis (*Figure 3C*). Second, $G\beta\gamma$ binding is cooperative with a factor of 0.30, which means the fourth $G\beta\gamma$ subunit binds with an affinity 37 times higher than the first. Attempts to fit the data imposing no cooperativity ($b$ = 1) yields higher residuals (0.126 compared to 0.064 when allowing cooperativity) and fail to replicate the steep rise in channel activity as a function of $G\beta\gamma$ concentration (*Figure 3—figure supplement 2*). The strong cooperative binding of four $G\beta\gamma$ subunits accounts for the steep sigmoidal dependence of GIRK current on membrane $G\beta\gamma$ concentration (*Figure 3A,B*). Third, $G\beta\gamma$ binds with a $Na^+$ cross cooperativity factor ($\eta$) of 0.63, which means $G\beta\gamma$ binds with 6-fold higher affinity when four $Na^+$ sites are occupied compared to when the $Na^+$ sites are not occupied. This effect of $Na^+$ on $G\beta\gamma$ affinity accounts for channel opening at lower membrane $G\beta\gamma$ concentrations as $Na^+$ concentration increases (*Figure 3B*).

The ability of $Na^+$ to increase the affinity of $G\beta\gamma$ is demonstrable in another way, through a simple, intuitive analysis. The family of data points (*Figure 3B*) conform well to the Hill equation with a single global Hill coefficient (n ≈ 3) but variable, $Na^+$-dependent equilibrium constant for $G\beta\gamma$ binding (*Figure 4A*). The $G\beta\gamma$ equilibrium constant decreases (i.e. the affinity for $G\beta\gamma$ increases) as $Na^+$ concentration increases according to a rectangular hyperbola, with a ~6-fold difference between maximum and minimum values (*Figure 4B*). The apparent equilibrium constant for the effect of $Na^+$ on $G\beta\gamma$ activation is ~5 mM, which is very close to the physiological $Na^+$ concentration in the cytoplasm of a resting neuron (*Rose and Ransom, 1997*). Thus, the GIRK2 channel's response to $G\beta\gamma$ should be sensitive to changes in $Na^+$ concentration right in the physiological range.

## Structural basis of $G\beta\gamma$ cooperativity and $Na^+$ activation

The atomic structures of the GIRK2 channel and its complex with ligands offers clues to the mechanistic underpinnings of $G\beta\gamma$ and $Na^+$ regulation of GIRK2 beyond the 4:1 stoichiometry of ligand binding (*Figure 5A*). When four $G\beta\gamma$ subunits bind to the cytoplasmic domain of GIRK2, which forms a ring made by the four channel subunits, they cause the ring to rotate as a rigid body with respect

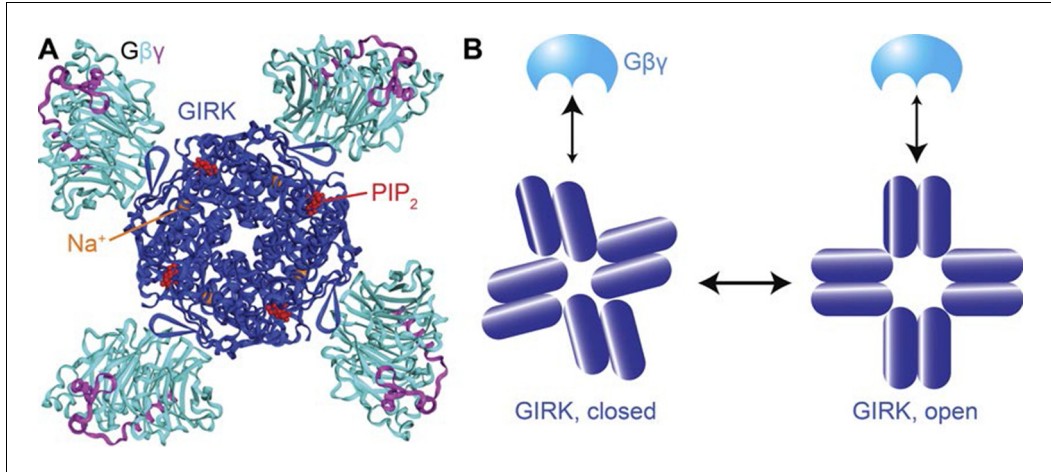

**Figure 5.** Structural basis for cooperativity in Gβγ activation of GIRK. (**A**) Top view of the atomic structure of GIRK2 in complex with Gβγ, Na$^+$ and PIP$_2$ (PDB ID: 4KFM). Four Gβγ, Na$^+$ and PIP$_2$ molcules bind to one GIRK2 homo-tetramer, associated with rotation of the cytoplasmic domain with respect to the transmembrane as the channel opens. (**B**) Gβγ binding favors the cytoplasmic-domain-rotated, open conformation of the channel. The rigid body rotation of the cytoplasmic domain is associated with increased affinity of four Gβγ binding sites simultaneously, giving rise to strong positive cooperativity.

to the pore, which twists open the helical bundle that forms the pore's gate (*Whorton et al., 2013*). Because the rigid body rotation involves all four subunits at once, conformational changes induced by the binding of Gβγ to one site will favor binding at the neighboring sites (i.e. positive cooperativity). A cartoon illustrating this concept depicts Gβγ binding more favorably to the channel's open conformation (*Figure 5B*). Because opening involves a concerted rotation of the subunits, all four Gβγ binding sites change to higher affinity at once, giving rise to strong positive cooperativity.

In addition to increasing Gβγ affinity, Na$^+$ also increases the GIRK2 current when the Gβγ binding sites are fully occupied: at the highest (0.03) Ni-NTA lipid mole fraction Na$^+$ increases current approximately 2.5-fold when Na$^+$ is increased from 0 mM to 32 mM (*Figure 3A*). This increase follows a rectangular hyperbola. The simplest physical explanation for this behavior, which is also consistent with the equilibrium model, is that Na$^+$ stabilizes the open, conductive state of the channel in direct proportion to its occupancy on the channel. In other words, four Gβγ subunits bind to GIRK2 and permit opening to a probability that is higher in proportion to occupancy of the Na$^+$ sites. Thus, by thermodynamic linkage, Na$^+$ would increase the apparent affinity of Gβγ for GIRK2 and it would also increase the maximum level of current reached when four Gβγ subunits bind. This physical mechanism is consistent with the location of the Na$^+$ binding sites at the interface between the cytoplasmic domains and the transmembrane pore, where opening is transduced through the binding of Gβγ (*Figure 5A*) (*Whorton et al., 2013*). It is also consistent with the observations that Na$^+$ in the absence of Gβγ does not open GIRK (*Figure 3A*) and in crystal structures Na$^+$ binds to GIRK but does not cause a rotation of the cytoplasmic domain in the absence of Gβγ (*Whorton and MacKinnon, 2011*). Thus, Na$^+$ facilitates Gβγ-mediated pore opening.

## Physiological role of Na$^+$ amplified Gβγ activation

How might neuronal electrical signaling be affected by the GIRK2 channel's dual regulation by Gβγ and Na$^+$? GIRK2 suppresses electrical activity in neurons when inhibitory neurotransmitters stimulate GPCRs on the cell surface, such as GABA$_B$ receptors, which release Gβγ on the intracellular membrane surface to open GIRK2 channels. At the same time, the level of GIRK2 channel opening – and therefore the level of neuronal inhibition – brought about by the released Gβγ potentially depends on the intracellular Na$^+$ concentration. This conclusion derives from the family of Gβγ activation curves (*Figure 3B*): at all Gβγ concentrations, the level of GIRK2 current is increased as Na$^+$ is increased. We refer to this phenomenon as Na$^+$-amplification of Gβγ-activated current. Na$^+$-amplification is clearly not constant but is instead a function of the Gβγ concentration: at high Gβγ

concentrations (right side of graph) amplification is 2.5-fold (i.e. GIRK2 current increases 2.5-fold) when $Na^+$ is increased from 0 to 32 mM, while at lower $G\beta\gamma$ concentrations (corresponding to the steep sigmoidal rise in current) amplification approaches ten fold. The potential importance of $Na^+$ amplification to neuronal electrical signaling lies in the fact that cytosolic $Na^+$ increases with higher levels of electrical activity due to $Na^+$ entry through both synaptic channels and voltage-dependent $Na^+$ channels (*Lasser-Ross and Ross, 1992*; *Fleidervish et al., 2010*, *Rose and Konnerth, 2001*). Thus, $Na^+$ amplification should, in principle, provide a mechanism for strengthening an inhibitory input to a more active neuron.

How large is $Na^+$-amplification in neurons? As noted above, the magnitude of amplification depends on the concentrations of $G\beta\gamma$ generated inside a neuron when its GPCRs are stimulated. While the concentration of $G\beta\gamma$ inside cells is unknown, the data in this study provide an approach to estimate its value. The rationale is as follows: the curves in *Figure 3B* characterize the amplification as a function of $G\beta\gamma$ concentration, therefore we should be able to solve the inverse problem of deducing $G\beta\gamma$ concentration by measuring the $Na^+$ amplification in a cell. *Figure 6* shows this analysis applied to GIRK currents recorded in midbrain dopamine neurons when baclofen was used to stimulate $GABA_B$ receptors. A recording pipette was used to set the cytoplasmic $Na^+$ concentration to either 0 or 27 mM. Baclofen-activated currents had the strongly inwardly-rectifying current-voltage relationship expected from GIRK channel activation (*Figure 6A*). Baclofen-activated GIRK current was much smaller in neurons recorded with 0 mM internal $Na^+$ compared with those recorded with 27 mM internal $Na^+$ (*Figure 6B,C*). Currents measured with 27 mM internal $Na^+$ were amplified by an average of 8 fold compared to currents with 0 mM $Na^+$. From the $G\beta\gamma/Na^+$ titration data (*Figure 3B*), 8-fold amplification corresponds to a $G\beta\gamma$ concentration of about 0.003 in NTA-lipid mole fraction units (*Figure 6D*). In this concentration range the amplification curve becomes very steep, allowing relatively small cell-to-cell variations in $G\beta\gamma$ concentration to translate into larger differences in current response (*Figure 6E*). This property offers an explanation for the large spread of current values measured upon baclofen activation in the presence of 27 mM $Na^+$. Most importantly, the estimated $G\beta\gamma$ concentration stimulated by baclofen in dopamine neurons (*Figure 6D,E*) is centered in the middle of the steep sigmoidal rising phase of the $G\beta\gamma$-activation curves (*Figure 3B*). In this regime even modest changes in intracellular $Na^+$ concentration should amplify $G\beta\gamma$-mediated inhibition of neuronal electrical activity. The intracellular $Na^+$ concentration in neurons is subject to complex regulation by multiple channels and transporters and changes during neuronal activity (*Rose and Ransom, 1997*). Intracellular $Na^+$ in dendrites can double during synaptic activity (*Rose and Konnerth, 2001*), and high local increases also occur in cell bodies and axons during action potential firing (*Lasser-Ross and Ross, 1992*; *Fleidervish et al., 2010*). Thus, the $Na^+$ amplification of GIRK currents likely occurs during normal physiological activity. Even stronger amplification is likely during epileptiform activity, when intracellular $Na^+$ can likely reach 30 mM (*Raimondo et al., 2015*).

## $G\beta\gamma$ membrane density and channel affinity

Taking into account the surface area of a lipid head group and the stoichiometry of 3 Ni-NTA lipid molecules per $sG\beta\gamma$-His10 subunit, a mole fraction value 0.003 (i.e. the concentration of $G\beta\gamma$ subunits estimated in dopamine neurons) translates into approximately 1200 $G\beta\gamma$ subunits per $\mu m^2$ of membrane. To place this 2-dimensional membrane density into more familiar concentration units we multiply the membrane surface area by the linear dimension of a $G\beta\gamma$ subunit (about 70 Å) to approximate a $G\beta\gamma$ concentration in the solution layer adjacent to the membrane equal to 280 $\mu$M. At this $G\beta\gamma$ concentration GIRK is between 10% and 80% activated, depending on the $Na^+$ concentration (*Figure 3B*). Thus, the apparent affinity of $G\beta\gamma$ for the GIRK channel is in this range.

This estimate is close to the affinity reported using ITC to study the interaction of lipid anchor-removed $G\beta\gamma$ in solution with the soluble cytoplasmic domain of a GIRK channel (250 $\mu$M) (*Yokogawa et al., 2011*). A more careful comparison, however, reveals a fascinating difference. Removed from the pore, the cytoplasmic domain, even though it is a tetramer with four $G\beta\gamma$ binding sites like the full channel, binds to $G\beta\gamma$ according to a 1:1 binding isotherm. As we have shown here, the full GIRK2 channel by contrast exhibits strong cooperativity, the first $G\beta\gamma$ subunit binding with very low affinity (equilibrium constant 0.019 mol fraction corresponding to 1.9 mM in the solution layer adjacent to the membrane) and the fourth binding with higher affinity (equilibrium constant $(0.019) \times (0.3)^3$ mol fraction corresponding to 50 $\mu$M in the solution layer adjacent to the

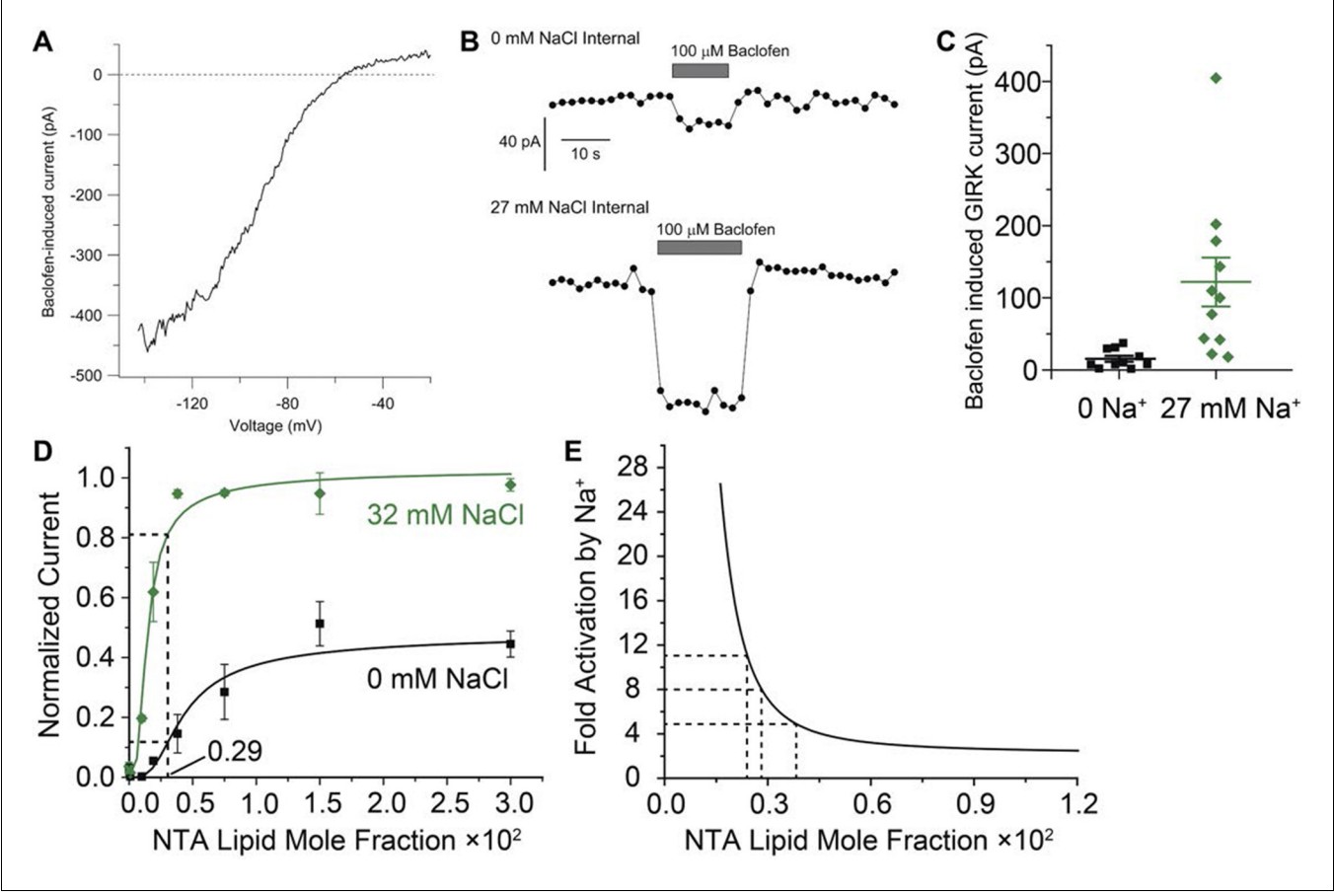

**Figure 6.** Estimation of Gβγ membrane density generated during GABA$_B$ receptor activation in mouse dopamine neurons. (**A**) Current-voltage relationship for current induced by 100 μM baclofen in a dissociated dopamine neuron from the mouse substantia nigra pars compacta recorded with an external solution containing 16 mM K$^+$ and an internal solution containing 126 mM K$^+$ and 27 mM Na$^+$. (**B**) Time-course of baclofen-induced current in two neurons recorded with an internal solution containing 0 mM Na$^+$ (top) or 27 mM Na$^+$ (bottom). The same scale is used for both recordings. Current was measured as the average current between -142 mV and -147 mV evoked by voltage ramps (1 mV/ms) from +8 to -147 mV delivered from a steady holding potential of -92 mV every 2 s. (**C**) Collected values for baclofen-induced GIRK current (mean ± SEM) in dopamine neurons equilibrated with 0 mM (n = 10) and 27 mM (n = 11) intracellular Na$^+$. (**D**) Data and curves from **Figure 3B** are used to estimate the concentration of Gβγ required to yield the 8-fold amplification of GIRK current observed in (**C**). (**E**) A Na$^+$ amplification curve is defined as the green curve (32 mM Na$^+$, which is near 27 mM) divided by the black curve (0 mM Na$^+$) in (**D**). Amplification is a steep function of Gβγ concentration near the stimulated levels of Gβγ in dopamine neurons.

membrane). This cooperativity, which gives rise to the steep dependence of GIRK channel activity on Gβγ concentration, is completely lost when the transmembrane pore is removed. The mechanism proposed for coupling Gβγ binding to pore opening, illustrated in **Figure 5**, offers an explanation: when the pore is removed, Gβγ binding free energy is no longer utilized to twist open the pore's helical gate, and at the same time the rotational origin of cooperativity disappears. The ITC-determined affinity of 250 μM lies in between the affinities of the first (1.9 mM) and fourth (50 μM) Gβγ subunits to bind to the intact, cooperative system.

In the context of other known protein complexes, the interaction of Gβγ with GIRK2 is weak, consistent with a short lifetime for the complex. For example, two proteins with a diffusion-limited association rate constant of, say, $10^7$ M$^{-1}$sec$^{-1}$, will remain in complex on average for less than 2 milliseconds if the equilibrium constant is 50 μM (i.e. affinity of the fourth Gβγ subunit) and the lifetime of an activated channel (GIRK2 with 4 Gβγ subunits, any one of which can dissociate) less than 0.5 milliseconds. Even if the association rate constant is smaller, the lifetime of an active channel will be brief compared to the duration of macroscopic GIRK current in a cell during GPCR stimulation (~1 s) (**Ford et al., 2009**). This means Gβγ apparently associates and dissociates many times on and

off the channel during a period of stimulation. Because Gβγ binds to Gα(GDP) with greater than ten thousand times higher affinity ($K_d \sim 1$ nM *Sarvazyan et al., 1998*) than to the channel, whenever Gα (GTP) hydrolyses GTP to GDP it will rapidly bind free Gβγ and remove it from the channel by mass action. Thus, the very weak binding of Gβγ to the channel means that the duration of GIRK current activation during GPCR stimulation will be controlled by the lifetime of Gα(GTP).

The Gβγ concentrations reported here represent the thermodynamic activity concentrations in equilibrium with the GIRK channel. In living cells it is distinctly possible that GIRK and GPCRs/G proteins reside in specialized regions of the cell membrane. In this case the relevant density of Gβγ is the local density near GIRK channels, which would be much higher than that averaged over the entire membrane. Such specialized regions would promote locally high Gβγ densities, in line with the relatively low affinity for GIRK subunits that allows rapid control of free Gβγ by Gα.

## Summary

A method to control the concentration (density) of G protein subunits in lipid membranes has let us reach the following conclusions. (1) Four Gβγ subunits bind to the GIRK channel with high cooperativity to give rise to a steep dependence of channel activity on membrane Gβγ concentration. (2) Intracellular $Na^+$ concentration increases Gβγ affinity with an apparent $K_{d-Na+}$ near the cytoplasmic $Na^+$ concentration of a resting neuron. (3) Inhibitory GPCR stimulation generates membrane Gβγ concentrations corresponding to the steep regime of the Gβγ-activation curve. (4) Properties (1) – (3) give rise to $Na^+$ amplification of Gβγ-activation. Such amplification provides a mechanism for strengthening GPCR inhibition when $Na^+$ enters neurons during activity. (5) Gβγ binds to GIRK with low affinity. Rapid equilibrium between Gβγ and GIRK allows rapid signal termination when Gα hydrolyses GTP to GDP.

## Materials and Methods

### Protein expression and purification

Mouse GIRK2 (residues 52–380) was expressed in *Pichia pastoris* and purified as previously described (*Whorton et al., 2011*). High Five (Life Technologies, Grand Island, NY) insect cells were infected with baculovirus bearing Human G protein subunits $\beta_1$ and $\gamma_2$. The G protein Gβγ subunit was then purified using an established protocol (*Whorton et al., 2013*; *Wang et al., 2014*). To produce non-lipid modified and His-tagged Gβγ, baculovirus bearing a mutant $G\gamma_2$ DNA with a C68S mutation and a 4- or 10-His tag connected to the C-terminus by a GSSG linker was generated. This mutant virus and the virus bearing $\beta_1$ DNA were co-infected into High Five cells. The purification process of non-lipid modified and His-tagged Gβγ is essentially the same as non-lipid modified Gβγ (*Wang et al., 2014*) except that PreScission protease digestion was not necessary since no cleavable tag was used.

### Reconstitution of proteoliposomes

GIRK2 proteoliposomes were reconstituted using a lipid mixture composed of 3:1 (w:w) 1-palmitoyl-2-oleoyl-sn-glycero-3-phosphoethanolamine (POPE): 1-palmitoyl-2-oleoyl-sn-glycero-3-phospho-(1'-rac-glycerol) (POPG) supplemented with 3%, 1.5%, 0.75%, 0.38%, 0.19%, 0.1%, 0.01% and 0% of DOGS-NTA-$Ni^{2+}$ lipid (Avanti, Alabaster, AL). 3:1 POPE:POPG was used for Gβγ reconstitution. The reconstitution protocol is essentially the same as previously described (*Wang et al., 2014*).

### Fluorescent labeling of sGβγ-His10 protein

Purified sGβγ-His10 protein was exchanged into conjugation buffer (50 mM potassium phosphate pH 7.4, 100 mM NaCl, 0.1 mM TCEP) and diluted to $\sim 1$ mg/ml. 5-fold molar excess of Alexa-Fluor 488 maleimide was mixed with the protein. The mixture was rotated at 4°C overnight. Labeled protein was affinity purified using $Ni^{2+}$-NTA (Qiagen, Valencia, CA) beads followed by size exclusion chromatography in a buffer containing 10 mM potassium phosphate pH 7.4 and 150 mM KCl. The labeling efficiency was approximately one dye per sGβγ-His10 protein.

## Confocal microscopy of giant unilamellar vesicles

DOPE: POPC 1: 1 (w: w) lipid mixture supplemented with 3% DOGS-NTA-Ni$^{2+}$ was used to produce GUVs. A few microliters of 1 mg/ml of the lipid mixture in chloroform were dried under vacuum on an electrically conductive Indium tin oxide coated glass slide (Sigma, St. Louis, MO) and electro-formed (*Meleard et al., 2009*) in a buffer containing 5 mM sodium phosphate with 300 mM sucrose (pH 7.4). After electroformation, the solution containing GUVs was diluted 5 fold into a buffer containing 10 mM potassium phosphate pH 7.4, 150 mM KCl and 2 nM NiSO$_4$. Fluorescently labeled sGβγ-His10 was then added to a final concentration of 2 μM, 1 μM, 0.5 μM, 0.25 μM, 0.125 μM and 63 nM. The equator plane of the GUVs was then imaged using a Leica DMI 6000 microscope controlled by Leica Application Suite X software (Leica, Buffalo Grove, IL). An oil immersion 63x objective (numerical aperture 1.40) was used. Fluorophore was excited with a white light laser positioned at 491 nm with a pinhole size of 1 airy unit, giving rise to a confocal plane thickness of about 360 nm. Emission light above 505 nm was detected with a Hyd detector in photon counting mode, which exhibited good linearity. A 3x 'smart zoom' was used. 1024 × 1024-pixel 8-bit depth images were recorded with 4x line averaging to increase signal to noise ratio. Each pixel corresponds to ~60 nm distance in x and y directions. Microscope and software settings were the same for all images acquired. The images were converted into tiff format using software Imaris. A Mathematica (Wolfram Research, Champaign, IL) script was used to quantify fluorescence from GUVs. A circle fitting algorithm was first performed to locate the GUV image. Fluorescence intensity in the center of the GUV was treated as background since no fluorophore should be present in this region (*Figure 2—figure supplement 1A*). The intensity of a 40-pixel (~2.4 μm) shell area around the fitted circle was integrated. This integrated value, divided by the circumference of the circle, is taken as the GUV fluorescence intensity. Images with saturated pixels in the quantification area were discarded. Measurements were replicated on 5–7 GUVs at each sGβγ-His10 concentration. Mean and SEM values were calculated. Data fitting was performed with Origin (OriginLab, Northampton, MA).

## Electrophysiology

DOPE: POPC 1: 1 (w: w) lipid mixtures supplemented with 3%, 1.5%, 0.75%, 0.38%, 0.19%, 0.1%, 0.01% and 0% of DOGS-NTA-Ni$^{2+}$ lipid were used to make planar bilayer lipid membranes across a ~100 micron diameter hole on a plastic transparency (*Montal and Mueller, 1972*; *Miller and Racker, 1976*; *Wang et al., 2014*). Buffer contained 10 mM potassium phosphate pH 8.2, 150 mM KCl on both sides of the membrane. 2 nM NiSO$_4$ and 2 mM MgCl$_2$ were also included in the top chamber. A detailed outline of the experimental procedure is illustrated in *Figure 1—figure supplement 1*. To measure the GIRK activity at a specific density of Gβγ in the membrane, a lipid bilayer containing a certain mole fraction of DOGS-NTA-Ni$^{2+}$ lipid was used for making the planar lipid bilayer. GIRK2 proteoliposomes with the same mole fraction of DOGS-NTA-Ni$^{2+}$ were subsequently fused into the lipid bilayer (*Figure 1—figure supplement 1A*). We use high KCl concentrations to facilitate vesicle fusion in our experiments. After the application of vesicles, 1 M KCl solution was applied at the membrane to complete the fusion process (*Figure 1—figure supplement 1B*). Since reducing reagent DTT and divalent metal chelator EDTA were present in the GIRK2 proteoliposomes reconstitution to preserve channel activity, Ni$^{2+}$ will not be present on the NTA lipid in these vesicles. To ensure that all NTA lipids are charged with Ni$^{2+}$, 1 mM NiSO$_4$ solution was applied at the membrane (*Figure 1—figure supplement 1C*). Then 2 μM sGβγ-His10 and 32 μM 1,2-dioctanoyl-sn-glycero-3-phospho-(1'-myo-inositol-4',5'-bisphosphate) (C8-PIP$_2$) were added to one side of the membrane. Given the chamber volume and membrane area, the molar ratio of sGβγ-His10 to DOGS-NTA-Ni$^{2+}$ always exceeded 10,000, thus ensuring a constant concentration of sGβγ-His10 except in specific experiments to study channel activation as a function of sGβγ-His10 concentration at a fixed DOGS-NTA-Ni$^{2+}$ lipid mole fraction (e.g. *Figure 2A*). Note that the C8-PIP$_2$ is maintained constant at 32 μM during all experiments. Only GIRK channels with their cytoplasmic surface facing the solution chamber containing C8-PIP$_2$ can be opened (*Figure 1—figure supplement 1D*) (*Wang et al., 2014*). A Na$^+$ titration was then followed (*Figure 1—figure supplement 1E*). Native (lipid-modified) Gβγ in the form of proteoliposomes was then fused to the membrane to saturate the Gβγ binding sites on GIRK, maximizing GIRK activation (*Figure 1—figure supplement 1F and G*). The maximized current level was used to normalize recordings for each membrane. The titration of sGβγ-His10 and sGβγ-His4 to a membrane containing 0.0019 mol fraction (*Figure 2A*) of Ni-NTA

lipid was performed by successive addition of the proteins to final concentrations of 0, 0.13, 0.25, 0.5, 1.0, 2.0 µM for sGβγ-His10 and 0, 0.25, 1.0, 2.0, 4.0 and 8.0 µM for sGβγ-His4. Native Gβγ was then used to saturate Gβγ binding. The analog signal was low-pass filtered at 1 kHz (Bessel) and digitized at 20 kHz with a Digidata 1322A or 1440A digitizer and recorded on a computer using the software suite pClamp (Molecular Devices, Sunnyvale, CA). Data was fitted with Origin (OriginLab, Northampton, MA). Measurements were replicated on 3–5 membranes and average and SEM values were calculated for each data point.

## Dopamine neuron electrophysiology

Dissociated dopamine neurons from the substantia nigra pars compacta were prepared from 13- to 19-day-old mice (*Kimm and Bean, 2014*) and studied with whole-cell patch clamp recording. GIRK current was evoked by application of 100 µM baclofen using an external solution containing 16 mM KCl, 139 mM NaCl, 1.5 mM $CaCl_2$, 1 mM $MgCl_2$, 13 mM glucose, 10 mM HEPES, pH adjusted to 7.4 with NaOH. The 0 $Na^+$ internal solution contained 140 mM K-gluconate, 13.5 mM NMDG-Cl, 1.6 mM $MgCl_2$, 0.09 mM EGTA, 4 mM MgATP, 14 mM creatine phosphate (Tris salt), 0.3 mM GTP (Tris salt), 9 mM HEPES, pH 7.4. The 27 mM $Na^+$ internal solution contained 13.5 mM NaCl, 13.5 mM Na-gluconate, 126 mM K-gluconate, 13.5 mM NMDG-Cl, 1.6 mM $MgCl_2$, 0.09 mM EGTA, 4 mM MgATP, 14 mM creatine phosphate (Tris salt), 0.3 mM GTP (Tris salt), 9 mM HEPES, pH 7.4. Pipette resistances were 2.5–3.5 MOhm. Whole-cell current was recorded during voltage ramps (1 mV/ms) from +8 to -147 mV delivered from a steady holding potential of -80 mV every 2 s. Baclofen-induced current was measured after 9–11 min of cell dialysis to allow equilibration with the internal solution. Baclofen-induced current was measured by averaging the current between -142 to -147 mV over ramps delivered during a 10-sec application of baclofen, subtracting the current before application of baclofen. Baclofen was applied in <1 s by moving the cell between a pair of quartz fiber flow pipes (250 µm internal diameter, 350 µm external diameter) glued onto an aluminum rod whose temperature was controlled by resistive heating elements and a feedback-controlled temperature controller (TC-344B, Warner Instruments, Hamden, CT). Recordings were made at 37°C.

## Basis of limiting slope analysis

For GIRK channels with 4 Gβγ binding sites, channel activity as a function of Gβγ concentration can be expressed as:

$$A = \frac{b^6 K_{db}{}^4 \theta_0 + 4 b^6 K_{db}{}^3 m \theta_1 + 6 b^5 K_{db}{}^2 m^2 \theta_2 + 4 b^3 K_{db} m^3 \theta_3 + m^4 \theta_4}{b^6 K_{db}{}^4 + 4 b^6 K_{db}{}^3 m + 6 b^5 K_{db}{}^2 m^2 + 4 b^3 K_{db} m^3 + m^4}, \tag{1}$$

where $K_{db}$ is the equilibrium dissociation constant for Gβγ binding to the first binding site, $b$ is the cooperativity factor for each successive Gβγ binding , $m$ is the Gβγ concentration and $\theta_j$ is the channel activity with $j$ Gβγ bound. If 4 Gβγ subunits are required to open GIRK ($\theta_j$ = 0 when j ≠ 4, $\theta_4$ = 1), then

$$\begin{aligned} \mathrm{Ln}(A) &= 4 \ln(m) - \ln\left(\frac{1}{b^6 K_{db}{}^4 + 4 b^6 K_{db}{}^3 m + 6 b^5 K_{db}{}^2 m^2 + 4 b^3 K_{db} m^3 + m^4}\right) \\ &\approx 4 \ln(m) - 4 \ln\left(\frac{1}{b^{3/2} K_{db}}\right) \end{aligned} \tag{2}$$

when $m \ll s \times K_{db}$. $s$ is the smallest of $4^{-1}$, $(b/6)^{1/2}$, $4^{-1/3} b$ and $b^{3/2}$. Equation (2) indicates that at low enough Gβγ concentrations a log-log plot of channel activity $A$ will be a linear function with a slope of 4. Moreover, if 3 bound Gβγ subunits are sufficient to activate the channel the slope will be less than 4. Thus, a log-log plot will reveal the number of Gβγ subunits required to open the channel if the slope can be measured at sufficiently low concentrations of Gβγ.

## Acknowledgements

We thank Matthew Whorton, Xiao Tao and Emily Brown for advice on biochemistry and reconstitution of the channels; Yi Chun Hsiung for assistance with insect cell culture; members of the MacKinnon laboratory for helpful discussions and Jue Chen for comments on the manuscript. This work was

supported in part by NIHGM43949 and NIHNS036855. RM is an investigator in the Howard Hughes Medical Institute.

## Additional information

### Funding

| Funder | Grant reference number | Author |
| --- | --- | --- |
| National Institutes of Health | NIHNS036855 | Bruce P Bean |
| National Institutes of Health | NIHGM43949 | Roderick MacKinnon |
| Howard Hughes Medical Institute | | Roderick MacKinnon |

The funders had no role in study design, data collection and interpretation, or the decision to submit the work for publication.

### Author contributions

WW, Designed the study, Collected electrophysiology data in planar lipid membranes, Collected data in GUV experiments, Analyzed data, Wrote the paper; KKT, Collected electrophysiology data in planar lipid membranes, Analyzed data; KW, Performed dopamine neuron electrophysiology and analyzed the data; BPB, Designed the study, Performed dopamine neuron electrophysiology and analyzed the data; RM, Designed the study, Analyzed data, Wrote the paper

### Author ORCIDs

Roderick MacKinnon, http://orcid.org/0000-0001-7605-4679

### Ethics

Animal experimentation: This study was performed in strict accordance with the recommendations in the Guide for the Care and Use of Laboratory Animals of the National Institutes of Health. All of the animals were handled according to a protocol approved by the institutional animal care and use committee (IACUC) of Harvard Medical School (Protocol #02538-R98).

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
