## [Decision Letter]

Thank you for submitting your article "Cooperative regulation by G proteins and Na^+^ of neuronal GIRK2 K^+^ channels" for consideration by *eLife*. Your article has been favorably evaluated by Richard Aldrich (Senior editor) and three reviewers, one of whom, Kenton Swartz, is a member of our Board of Reviewing Editors. The other two reviewers have agreed to reveal their identities: Chris Miller and Don Hilgemann.

The reviewers have discussed the reviews with one another and the Reviewing Editor has drafted this decision to help you prepare a revised submission.

Summary:

This substantively superb and well-written paper describes breakthrough experiments that for the first time paint a quantitative picture of Gβγ activation of GIRK channels and its modulation by Na^+^ ion. The authors develop a biochemically defined, reconstituted systems with all the works needed for a serious analysis: membrane-tethered Gβγ, PIP_2_ at fixed concentration, and purified GIRK2 homotetramers. They describe and verify an assay for systematically varying the membrane surface density of Gβγ in order to quantify its activation of GIRK2 macroscopic currents in planar lipid bilayers – something never convincingly achieved previously. They fit their results to a simple model in which all 4 Gβγ subunits must bind to GIRK before activation can occur, also showing that Gβγ binding to the successive subunits involves a roughly 3-fold (kT's worth) positive cooperativity leading to a persistent steepness in the switch-like activation curve. The interplay between Gβγ and Na^+^ binding is also derived from these reduced-system experiments. And then this is all plausibly related to the situation in dopamine neurons. It's a real tour-de-force that opens the way for many more such analyses of G-protein activation of ion channels.

Requested revisions:

Despite the overall excellence of the paper, we think that a few places that need shoring up.

1) Figure 1. This is the key raw data showing the assay for activation of GIRK2 currents. The short, 10-second segments look flat, as though they are at equilibrium, but this is meaningful only if we'd have a sense of the time required to obtain these effects of adding 2 μm Gβγ and of the Gβγ in liposomes fused into the bilayer. Are these really representative of equilibrium levels? Does it take minutes or second in the "breaks" on the figure? In Figure 2, were a range of Gβγ concentrations tested in each experiment? In Figure 3, were a range of Na concentrations tested in each recording? Illustrative time courses for the experiments would help the reader to understand how experiments were done. Also, it would be nice to add exemplary recordings to Figure 6 at the two Na concentrations tested.

2) Sigmoid curves of Figure 2, Figure 3, Figure 4. In deriving parameters from these sigmoid fits, we would be more convinced of the Gβγ cooperativity factors if an example were given showing that the data could not be well fit by a model in which no cooperativity is present. Such plots will still give a quartic 'foot' at low Gβγ (as you point out in eq (1) and (2)), and it might be hard to distinguish a coop factor of 0.3 from one of 1.0 from the rather sparse number of points on the sigmoid plots shown.

3) Concerning the conclusions expressed in the Abstract: the authors state that Na acts "mostly by increasing Gβγ affinity". Can the authors state in a few words why the outcome is 'mostly' and not entirely? Perhaps the authors could point to Figure 3 and discuss the increase in current with added Na^+^ at concentrations of NTA-lipids and Gβγ that are saturating. The Discussion could also be somewhat more clear about this. The authors should also consider including a direct measurement by single-channel recording of the value of maximum open probability in the absence and presence of Na^+^. That measurement would help to complete the picture of Na^+^ modulation, and should be relatively straightforward to do, given the excellent reproducibility of the bilayer macroscopic recordings.

4) Even in neurons, cytoplasmic Na may or may *not* increase with increase of firing frequency. If the firing patterns result in depolarization for substantial periods of time, compared to time spent in a repolarized state (as in cardiac muscle), cytoplasmic Na may go down, not up, with increased firing. Cytoplasmic Na is subject to many cellular influences besides excitation rate and pattern. This includes the activities of transporters and ligand gated channels that may have a larger influence in many places where these channels are expressed than the influence of action potential trains. Perhaps it is worth touching on some of this underlying complexity in the Discussion.

---

## [Author Response]

Requested revisions:

Despite the overall excellence of the paper, we think that a few places that need shoring up.

1) Figure 1. This is the key raw data showing the assay for activation of GIRK2 currents. The short, 10-second segments look flat, as though they are at equilibrium, but this is meaningful only if we'd have a sense of the time required to obtain these effects of adding 2 μm Gβγ and of the Gβγ in liposomes fused into the bilayer. Are these really representative of equilibrium levels? Does it take minutes or second in the "breaks" on the figure?

We have added Figure 1—figure supplement 2 to show example activation traces when 2 μM sGβγ-His10 (Figure 1—figure supplement 2) or lipid modified Gβγ in liposomes (Figure 1—figure supplement 2) are added to bilayers to activate GIRK channels. The sentence “Example current traces of activation by sGβγ-His10 and lipid modified Gβγ in liposomes are shown in Figure 1—figure supplement 2.” has been added to the Figure 1 legend.

It usually takes a few seconds to a few tens of seconds for sGβγ-His10 activation to reach equilibrium. Once equilibrated, current is stable for tens of minutes. Activation by Gβγ in liposomes is slower. Gβγ liposomes are applied multiple times until current does not increase further to ensure Gβγ binding is ‘saturated’. After saturation, current is stable for many minutes. The “breaks” in Figure 1 contain voltage families after each condition change and take around 4~8 minutes each. The traces shown in Figure 1 are taken from equilibrium levels before and after the “breaks”.

In Figure 2, were a range of Gβγ concentrations tested in each experiment?

This is correct. We have added the sentence “The titration of sGβγ-His10 and sGβγ-His4 to a membrane containing 0.0019 mole fraction (Figure 2) of Ni-NTA lipid was performed by successive addition of the proteins to final concentrations of 0, 0.13, 0.25, 0.5, 1.0, 2.0 μM for sGβγ-His10 and 0, 0.25, 1.0, 2.0, 4.0 and 8.0 μM for sGβγ-His4. Native Gβγ was then used to saturate Gβγ binding” to the Methods.

In Figure 3, were a range of Na^+^ concentrations tested in each recording? Illustrative time courses for the experiments would help the reader to understand how experiments were done.

The detailed process of the experiment is shown in Figure 1—figure supplement 1 and described in the Methods (subsection “Electrophysiology”). In short, Na^+^ titration (at concentrations of 0, 4 mM, 8 mM, 16 mM and 32 mM) was performed at each Ni-NTA lipid mole fraction. The currents were then normalized to the maximal activity by repetitive application of lipid-modified Gβγ in liposomes to saturate Gβγ binding.

Also, it would be nice to add exemplary recordings to Figure 6 at the two Na concentrations tested.

We have added examples of the current-voltage relationship for baclofen-induced current, showing that it exhibits the strongly inwardly-rectifying current-voltage relationship expected for GIRK current (Figure 6) as well as exemplar time-courses for the activation of current at each of the two Na^+^ concentrations (Figure 6).

*2) Sigmoid curves of Figure 2, Figure 3, Figure 4. In deriving parameters from these sigmoid fits, we would be more convinced of the Gβγ cooperativity factors if an example were given showing that the data could not be well fit by a model in which no cooperativity is present. Such plots will still give a quartic 'foot' at low Gβγ (as you point out in eq (1) and (2)), and it might be hard to distinguish a coop factor of 0.3 from one of 1.0 from the rather sparse number of points on the sigmoid plots shown.* The sigmoidal shape of dose-response curves could indeed arise from the requirement of multiple ligand binding for activation but not cooperativity in binding itself. However, the cooperativity affects the “steepness” of the Gβγ response. To demonstrate this, we added Figure 3—figure supplement 2 to compare iso-Na^+^ cross-sections of the model-predicted surface allowing or not allowing cooperativity. The curves not allowing cooperativity are not “steep” enough to follow the trend of the data points, exhibiting systematic deviation (obvious at Ni-NTA lipid mole fractions around 0.0038 and 0.03). The fit assuming no cooperativity gives rise to a higher scaled residual sum of squares of 0.126 compared to 0.064 for the fit allowing cooperativity. We have added a sentence “Attempts to fit the data assuming no cooperativity yield a higher residual (0.126 compared to 0.064 when allowing cooperativity) and exhibit systematic deviation from experimental data (Figure 3—figure supplement 2).” Also, the sentence “A comparison of fits to the data using cooperative and non-cooperative models is shown in Figure 3—figure supplement 2” has been added to the end of Figure 3 legend.

The equations (1) and (2) were derived assuming no cooperativity for simplicity. In light of your question, we have modified these equations to include the cooperativity factor “*b*” (subsection “Basis of limiting slope analysis”). When cooperativity is considered, the slope of the log-log plot (Figure 3) at sufficiently low Gβγ concentrations is still determined by the number of Gβγ required to open the channel. However, the condition of “sufficiently low” is shifted as described in the aforementioned Methods section.

*3) Concerning the conclusions expressed in the Abstract: the authors state that Na acts "mostly by increasing G β-γ affinity". Can the authors state in a few words why the outcome is 'mostly' and not entirely? Perhaps the authors could point to Figure 3 and discuss the increase in current with added* Na^+^*at concentrations of NTA-lipids and Gβγ that are saturating. The Discussion could also be somewhat more clear about this. The authors should also consider including a direct measurement by single-channel recording of the value of maximum open probability in the absence and presence of Na^+^. That measurement would help to complete the picture of* Na^+^
*modulation, and should be relatively straightforward to do, given the excellent reproducibility of the bilayer macroscopic recordings.* The further activation of GIRK by Na^+^ (~2.5 fold) in the presence of saturating concentrations of Gβγ is very interesting. It is exactly the reason why we concluded that Na^+^ acts mostly but not entirely by increasing Gβγ affinity (in which case at sufficiently high Gβγ concentrations Na^+^ would have no further effect). In the subsection “Structural basis of Gβγ cooperativity and Na+ activation” we have a paragraph discussing the possible mechanism underlying this phenomenon. We also have wondered how Na^+^ affects the open probability of the channel when all Gβγ sites are occupied. Single GIRK channels are notoriously spikey (very short open dwell time), making it difficult to get accurate data. Nevertheless, we had addressed this question using non-stationary noise analysis from the macroscopic current recordings. Figure 7 shows what we found graphing current variance as a function of mean current at different Na^+^ concentrations (in mM, 0, 2, 4, 8, 16, 32 from left to right):10.7554/eLife.15751.014Author Response Image 1.**DOI:**
http://dx.doi.org/10.7554/eLife.15751.014

The solid line corresponds to *σ^2^* = *i × I* – *I^2^ / N*, where *σ^2^* is the variance, *I* is the mean current, *i* is the unitary current and *N* is the number of channels in the membrane. The two arrows indicate data points at 0 mM and 32 mM NaCl. As you can see, even at 32 mM Na^+^ the open probability *P* (= *I / N* × *i)* is less than 0.5 (would occur at the maximum of the parabola). We did not include this analysis in the paper because our conclusions in the paper are independent of whatever this maximum open probability is. We do agree with you that it is nevertheless an interesting point. We wonder whether the spikey nature of GIRK channels (and the difficulty in approaching an open probability of 1.0) could be related to the low affinity of Gβγ. If 4 have to be bound and their affinity is low (rapid dissociation), it is easy to understand why the open dwell time would be very short. This of course is speculation but we are happy you asked.

4) Even in neurons, cytoplasmic Na may or may not increase with increase of firing frequency. If the firing patterns result in depolarization for substantial periods of time, compared to time spent in a repolarized state (as in cardiac muscle), cytoplasmic Na may go down, not up, with increased firing. Cytoplasmic Na is subject to many cellular influences besides excitation rate and pattern. This includes the activities of transporters and ligand gated channels that may have a larger influence in many places where these channels are expressed than the influence of action potential trains. Perhaps it is worth touching on some of this underlying complexity in the Discussion.

We have modified the discussion of this point to acknowledge the complexity of control of intracellular Na^+^ by multiple channels and transporters (subsection “Physiological role of Na+ amplified Gβγ activation”, second paragraph). That said, in all cases where it has been examined, intracellular sodium in neurons has been observed to increase with increases in either synaptic activity or action potential firing, and we have added references to more fully cite this work.